

Estimates of the organic aerosol volatility in a boreal forest using two
independent methods

Juan Hong[1], Mikko Äijälä[1], Silja A. K. Häme[1], Liqing Hao[2], Jonathan Duplissy[1,3],
Liine M. Heikkinen[1], Wei Nie[4], Jyri Mikkilä[1], Markku Kulmala[1], Annele Virtanen[2],
Mikael Ehn[1], Pauli Paasonen[1], Douglas R. Worsnop[5], Ilona Riipinen[6], Tuukka
Petäjä[1] and Veli-Matti Kerminen[1]

[1]Department of Physics, University of Helsinki, P.O. Box 64, 00014 Helsinki,
Finland
[2]Department of Applied Physics, University of Eastern Finland, Kuopio 70211,
Finland
[3]Helsinki Institute of Physics, University of Helsinki, P.O. Box 64, 00014 Helsinki,
Finland
[4]Institute for Climate and Global Change Research & School of Atmospheric
Sciences, Nanjing University, Nanjing, 210093, China
[5]Aerodyne Research, Inc., Billerica, Massachusetts, USA
[6]Department of Environmental Science and Analytical Chemistry, Stockholm
University, 10961 Stockholm, Sweden

Abstract

Volatility distribution of secondary organic aerosols, i.e. the particle mass
fractions of semi-volatile, low-volatility and extremely low-volatility organic
compounds was characterized in a boreal forest environment of Hyytiälä,
Southern Finland. This was done by interpreting field measurements using a
Volatility Tandem Differential Mobility Analyzer (VTDMA) with a kinetic
evaporation model. The field measurements were performed during April and
May of 2014. On average, 40 % of organics in particles was semi-volatile; 34 %
low-volatility organics and 26 % extremely low-volatility organics. The model
was, however, very sensitive towards the vaporization enthalpies assumed for
the organics ($\Delta H_{VAP}$). The best agreement between the observed and modeled
temperature-dependence of the evaporation was obtained when effective
vaporization enthalpy values of 80 kJ/mol were assumed. The low effective
enthalpy value might result from several potential reasons, including molecular
decomposition or dissociation that might occur in the particle phase upon
heating, mixture effects and compound-dependent uncertainties in the mass
accommodation coefficient. In addition to the VTDMA-based analysis, semi-
volatile and low-volatile organic mass fractions were independently determined
by applying Positive Matrix Factorization (PMF) to High-Resolution Aerosol
Mass Spectrometer (HR-AMS) data. The factor separation was based on the
oxygenation levels of organics, specifically the relative abundance of mass ions at
$m/z$ 43 ($f43$) and $m/z$ 44 ($f44$). The mass fractions of these two organic groups
were compared against the VTDMA-based results. In general, the agreement
between the VTDMA results and the PMF-derived mass fractions of organics was
reasonable with a linear correlation coefficient of around 0.4 with $\Delta H_{VAP}$ = 80
kJ/mol set for all organic groups. The prospect of determining of extremely low


volatile organic aerosols (ELVOA) from AMS data using the PMF analysis should be assessed in future studies.

## 1 Introduction

Atmospheric aerosols influence the Earth's climate both directly and indirectly through affecting the radiation balance, and altering the albedo, lifetime and precipitation patterns of clouds (IPCC, 2013). However, uncertainty in the spatial and temporal variability of the aerosol size distribution, chemical composition and physicochemical properties make it difficult to quantify the aerosol climate effects. The physicochemical properties of atmospheric aerosol populations vary (e.g. Jimenez et al., 2009). In terms of aerosol chemical composition measurements, one of the greatest challenges is the presence of a vast number of different organic components in the particles (Kanakidou et al., 2005; Goldstein et al., 2007; Kroll et al., 2011; Donahue et al., 2013). Understanding of the chemical and physical properties of these organic compounds remains incomplete (Hallquist et al., 2009).

One of the key physicochemical properties of atmospheric organic compounds is their volatility, which determines their partitioning between the gas and particle phase (Pankow et al., 1994; Bilde et al., 2015). Atmospheric aerosol particles are mixtures of organic and inorganic compounds having different volatilities. Volatilities of the common inorganic species are relatively well known, while information on the volatility of organic species, especially on extremely low-volatile organics (Ehn et al., 2014; Bilde et al., 2015), is still incomplete.

Different compounds evaporate differently at different temperatures depending on their volatilities, described with saturation vapor concentrations and enthalpies of vaporization (Kreidenweis et al., 1998). Therefore, measuring the evaporation of particles at different temperatures provides indirect information on the volatility of particles. Thermodenuders (TD) where particle populations are heated, often coupled with a Tandem Differential Mobility Analyzer (TDMA), are often used to obtain volatility information on particles. More quantitative information on the volatility distribution can be further obtained by coupling the measurement data with a kinetic evaporation model (e.g. Riipinen et al., 2010; Cappa et al., 2010) that describes the evaporation rate of aerosols inside the TD. While the combination of different TD-setups has been applied to quantify the volatility of laboratory-generated aerosol particles (e.g. Häkkinen et al., 2014) as well as field observations (e.g. Lee et al., 2010; Cappa et al., 2010; Häkkinen et al., 2012), it has not been utilized to determine the volatility distribution of ambient organic aerosol in a boreal environment.

Positive Matrix Factorization (PMF) is one of the widely used factor analysis techniques for environmental applications. PMF allows separating organic aerosol (OA) mass spectra into individual groups based on their bulk chemical characteristics, providing information on the OA sources and atmospheric processing (Lanz et al., 2007; Huffman et al., 2009; Zhang et al., 2011). Typical organic groups determined using the PMF analysis include hydrocarbon-like OA (HOA) from primary sources, biomass burning OA (BBOA) and oxygenated OA



(OOA) from secondary sources. OOA can be further separated into low volatility OOA (LV-OOA) and semi-volatile OOA (SV-OOA). Even though there have been multiple studies using PMF to identify different organic OA groups from ambient data (Ulbrich et al., 2009; Hildebrandt et al., 2010; Ng et al., 2010), especially the

SV-OOA and LV-OOA groups, to our knowledge there are only few studies (Cappa and Jimenez, 2010; Paciga et al., 2016) attempting to directly connect the oxygenation levels from these two OOA groups with the volatility of OA obtained by other methods. Comparing the volatility distribution obtained using a mass transfer model and VTDMA data to the oxidation level derived from the AMS

data using PMF can help in quantifying the volatilities of SV-OOA and LV-OOA.

In this study, we provide quantitative information on volatility distributions of organic species of ambient aerosol in a boreal forest environment. The sensitivity of the VTDMA-based analysis to assumed vaporization enthalpy

values of organics is discussed. The VTDMA-derived volatility distributions are compared with the ones obtained from the statistical analysis of the AMS.

## 2 Methods

### 2.1 Measurements site

The measurements were performed at the Hyytiälä SMEAR II (Station for Measuring Ecosystem-Atmosphere Relations II) between 14 April and 31May 2014. The SMEAR II station, located in Southern Finland, is surrounded by a 54-

year-old pine forest. The closest large city is Tampere with a population of around 213 000 and about 48 km to the South-West of the measurement station.

A series of ambient parameters, e.g., particle number size distribution of 3-1000 nm particles (Aalto et al., 2001), ambient meteorological conditions such as

temperature, relative humidity, solar radiation, wind speed and wind direction as well as gas phase concentrations of e.g. $SO_2$, $O_3$, $NO_X$, are continuously measured at the station.

### 2.2 Particle Volatility


The evaporation behavior of submicron aerosols was investigated using a Volatility Tandem Differential Mobility Analyzer (VTDMA), which is part of a Volatility-Hygroscopicity Tandem Differential Mobility Analyzer (VH-TDMA) system (Hong et al., 2014). A brief schematic view of the VTDMA is shown in Fig.

1. In brief, a monodisperse aerosol population (particle diameter of 30, 60, 100 and 145 nm; RH < 10%) was selected by a Hauke-type Differential Mobility Analyzer (DMA; Winklmayr et al., 1991). The aerosol flow was then heated by a thermodenuder at a set temperature, after which the remaining aerosol material was introduced into a second DMA followed by a condensation particle counter

(CPC, TSI 3010 & TSI 3772), where the number size distribution of the aerosol after heating was measured. The thermodenuder is a 50-cm stainless steel tube. No adsorptive material for removing the gas phase was used after the heating section. The residence time inside the thermodenuder was around 2.5 s. The



heating temperature of the setup ramped from 25 °C to 280 °C with a time
resolution of about an hour.

The VTDMA measures the particle diameter (and concentration) after heating at
each temperature for particles of certain initial size. From this information
volume fraction remaining (VFR) after the heating of particles of diameter $D_P$ can
be defined as follows

$$VFR(D_P) = \frac{D_p{}^3(T)}{D_p{}^3(T_{room})} = GF_V^3(T).$$                                          (1)

$GF_V$ describes how much of the particles shrink in size upon heating. With VFR =
1 particles are considered to be non-volatile, and with VFR = 0 particles fully
evaporate upon heating. The mass fraction remaining (MFR) after the heating
was assumed to be equivalent to VFR assuming that particle density was
constant upon heating (Häkkinen et al., 2012).

Data during a running time window (5 hours) was inserted into the model with a
time resolution of half an hour to make sure a full thermogram, i.e. the VFR or
MFR as a function of temperature, could be obtained. The corresponding results
represented the conditions (VFR or MFR) at the median time of the 5-hour time
window.

2.3 Particle chemical composition

A High-Resolution Aerosol Mass Spectrometer (HR-AMS, Aerodyne Research Inc.,
Billerica, USA) was used to determine the chemical composition of aerosol
particles during the experimental period. Detailed description of the instrument,
measurement and data processing can be found in other publications (DeCarlo et
al., 2006; Canagaratna et al., 2007). A Sunset semi-continuous OC/EC analyzer
was deployed to determine the mass concentrations of organic carbon (OC) and
elemental carbon (EC) concentrations in aerosols using a thermal-optical
protocol (Bauer et al., 2009).

2.3.1 Pairing of inorganic species


The neutral inorganic salts were calculated from the molar concentration of all
ions measured by the HR-AMS based on ion-pairing schemes introduced by
Reilly and Wood (1969) and Gysel et al. (2007). $SO_4^{2-}$ was first neutralized by
$NH_4^+$, and the excess of $NH_4^+$ was then used to neutralize $NO_3^-$. The simplified
ion-paring scheme was introduced as below:

$$n_{H_2SO_4} = max\big(0, n_{SO_4^{2-}} - n_{NH_4^+}\big),$$
$$n_{NH_4HSO_4} = min\big(2n_{SO_4^{2-}} - n_{NH_4^+}, n_{NH_4^+}\big),$$
$$n_{(NH_4)_2SO_4} = min(max\big(n_{NH_4^+} - n_{SO_4^{2-}}, 0\big), n_{SO_4^{2-}}),$$
$$n_{NH_4NO_3} = min\big(max\big(n_{NH_4^+} - 2n_{SO_4^{2-}}, 0\big), n_{NO_3^-}\big),$$                    (2)

where $n$ denotes the number of moles. This should naturally be treated only as a
rough estimation, as the scheme assumes perfectly internally mixed particles,





and the competing bonding of $NH_4^+$ between $SO_4^{2-}$ and $NO_3^-$ in particle phase is
not fully described.

### 2.3.2 Positive matrix factorization (PMF) of organic aerosol composition

Factor analysis is commonly used to de-convolve the time-dependent OA
concentrations and mass spectra into their basic components, based on a linear
algebraic model explaining the observed variance. The resulting components, i.e.
factors, are interpretable as separate organic sub-groups. The sum of these
organic groups' concentrations should closely match the measured organic
aerosol mass. Positive Matrix Factorization (Paatero et al., 1997) is one of these
component analysis techniques, constrained so that only positive concentration
and mass spectra are obtained. In this study, PMF was applied by using the PMF2
algorithm implemented with the user-interface Sofi by Canonaco et al. (2013) to
the organic aerosol data measured by the HR-AMS.

### 2.4 Kinetic evaporation model

A time-dependent evaporation model (Riipinen et al., 2010) was used to
simulate the evaporation of a monodisperse aerosol population in a heated flow
tube by solving the relevant mass transfer equations. The TD temperature profile,
residence time, initial particle size and the thermophysical properties of the
aerosol particles were used as input to the model. The volatility of the aerosol
constituents was described by the effective saturation concentration, $C^*$, at
standard conditions.

According to Donahue et al. (2013) and Murphy et al. (2014), compounds with
different effective saturation vapor concentrations can be classified into
extremely low volatile (ELVOC; $C^* < 10^{-4}$ µg/m³), low volatile (LVOC; $10^{-3}$ µg/m³
$< C^* < 10^{-1}$ µg/m³), semi-volatile (SVOC; $10^{-0.5}$ µg/m³ $< C^* < 10^{2.5}$ µg/m³) and
intermediate volatile (IVOC; $10^{2.5}$ µg/m³ $< C^* < 10^{6.5}$ µg/m³) organic compounds.
In the model, we assume the OA to consist of three organic groups with their
individual characteristic saturation concentration of $10^{-5}$ (ELVOA), $10^{-2}$ (LVOA)
and 10 µg/m³ (SVOA), corresponding to $10^{-10}$, $10^{-7}$, $10^{-5}$ Pa, or $10^4$, $10^7$, $10^{10}$
molec/cm³: the aim being to obtain the particle mass fractions of each of the
organic group. The ambient particles were assumed to be a mixture of six species,
including the afore mentioned organic groups and three inorganic components,
namely ammonium nitrate (AN), ammonium sulfate (AS) and elemental carbon
(EC). AN and AS were assigned with their own characteristic effective saturation
vapor concentration and effective vaporization enthalpies obtained from
laboratory measurements (see Table 1). Elemental carbon (EC) was assumed to
be non-volatile in the temperature range used in this study (assuming $C^*$ of $10^{-30}$
µg/m³). As a result, the corresponding average volatility distribution of the
ambient aerosol was obtained by letting the difference between the measured
and modeled evaporation of the ambient aerosol to reach a minimum with a
certain pair of mass fractions of these three organic groups together with known
mass fractions of AS, AN and EC from HR-AMS and OC/EC measurements. The
MATLAB optimization function *fmincon* with constrains was used to obtain the
optimal fit between the measured and modeled thermograms. This function was



constrained by setting the sum of mass fraction of organics from the model to equal the mass fraction of OA measured by HR-AMS.


The input parameters, including the physicochemical properties of the six components used for the model as well as particle properties, are summarized in Table 1. Specifically, a mass accommodation coefficient of unity was used along the whole study, thus yielding the maximum estimates for $C^*$s. To best match the overlapping size ranges of the instruments (VTDMA 30-145 nm and HR-AMS 60-1000 nm), in this study we focus on modeling the evaporation of 100 nm particles.

Lee et al. (2010) reported that the modeled MFR is likely to depend strongly on the vaporization enthalpy values. Hence, sensitivity tests towards this variable were performed. In the sensitivity analysis the vaporization enthalpy values of organics with different volatilities were either assumed to be the same or varied for the different organics, e.g. [100 80 60] kJ/mol. Epstein et al. (2010) fitted the average $\Delta H_{VAP}$ as a function of $log_{10}C^*$ to a set of surrogate organic compounds
and obtained the following relationship:

$$\Delta H_{VAP} = -11 * log_{10}C^* + 129. \tag{3}$$

where $\Delta H_{VAP}$ and $C^*$ are in the units kJ/mol and µg/m³, respectively. This $C^*$-dependent $\Delta H_{VAP}$ (Eq. 3) was also tested in the model caclulations. The
combinations of enthalpy values of all these three organic groups used in this study are summarized in Table 2.

3 Results and discussion

3.1 Inorganic volatility

Figure 2 illustrates the measured and model-interpreted thermograms (i.e. MFR as a function of the heating temperature) of ammonium nitrate and ammonium sulfate. Vallina et al. (2007) reported that for 150 nm AN and AS particles, the
volatilization temperatures (temperature of full particle evaporation) are around 60 °C and 180 °C, respectively, by using a similar VTDMA system with a residence time of around one second. According to the experimental curves (black line) in Fig. 2, AN and AS evaporated completely at around 45 °C and 180 °C, respectively. These results are close to those of Vallina et al. (2007), when the
effect of faster evaporation for smaller particles and longer residence time of this study are taken into account.

Modeled thermograms for both AN and AS were obtained by treating the saturation vapor pressures and enthalpy of vaporization as fitting parameters.
The optimum $C^*$-$\Delta H_{VAP}$ pair was obtained by minimizing the difference between the measured and model-interpreted thermograms (red lines in Fig. 2). The measured evaporation of AN was reproduced using $C^*$ and $\Delta H_{VAP}$ of 76 µg/m³ (corresponding to 2.6·10⁻³ Pa) and 152 kJ/mol, respectively. The obtained $\Delta H_{VAP}$ is 1.5 times higher than reported previously (Brandner et al., 1962; Hildenbrand
et al, 2010; Salo et al., 2011), and the saturation vapor concentration of the same





magnitude as in previous studies (Brandner et al., 1962; Chien et al., 2010). For AS, $C^*$ and $\Delta H_{VAP}$ of $2 \cdot 10^{-3}$ µg/m$^3$ and $\Delta H_{VAP}$ of 94 kJ/mol reproduced the measurements best. Chien et al. (2010) reported an observation of AN partially decomposing to NH$_3$ and HNO$_3$ upon heating. Huffman et al. (2009) similarly

suggested that AS might decompose to ammonium bisulfate and ammonia when heating to around 90-140 °C. The evaporation mechanisms of these inorganics might be different from the evaporation of organics, where the $C^*$-dependent $\Delta H_{VAP}$ was obtained (Epstein et al., 2010), since besides sublimation, decomposition might also occur during the evaporation of inorganics. Hence, the

vaporization enthalpy from Eq. 3 is not used for the simulation of the evaporation of inorganics. Instead, we selected the saturation vapor concentration and $\Delta H_{VAP}$ values as shown in red curve in Fig. 3 for both AN and AS during modeling analysis for all the cases below. Moreover, according to the saturation vapor concentration obtained for AN and AS in this study, we can

conclude that AN and AS can be considered as semi-volatile and low-volatility compounds, respectively.

The measured thermogram and corresponding evaporation mechanism of ammonium bisulfate (NH$_4$HSO$_4$) are not available at present. In order to neglect

the effect of ammonium bisulfate on particle evaporation behavior, only data with the mass fraction of ammonium bisulfate less than 10% of total aerosol mass (calculated from Eq. 2) was analyzed.

3.2 Performance of the model for TD data on the organic mixtures


Figure 3 shows example fits to the observed thermograms using different combinations of organic vaporization enthalpies (Table 2). The median norm of residuals, which describes the difference between the fit and observed thermograms, was the largest when the $C^*$ dependent $\Delta H_{VAP}$ values for organics

were applied in the model. As $\Delta H_{VAP}$ increases, the sensitivity of $C^*$ to temperature changes also increases, requiring also lower $C^*$ values to match observations (see the red curve in Fig. 3). This is in line with Cappa & Jimenez (2010) who suggested that value of $C^*$ as low as $10^{-15}$ µg/m$^3$ for extremely low volatility material is required to match the observations when $C^*$-dependent

vaporization enthalpy values of Epstein et al. (2010) are used.

The $C^*$-independent vaporization enthalpy values (e.g. Combinations 1 to 8 in Table 2), better agreement between the fitted and observed thermograms (Fig. 3) was obtained. Donahue et al. (2006) pointed out that artificially low $\Delta H_{VAP}$ values

are expected when we present the complex organic mixture aerosol with one single organic compound or of very few components. The artificially low $\Delta H_{VAP}$ values should thus be rather referred to effective enthalpy of vaporization (see e.g. Offenberg et al., 2006). According to the performance of the model to TD data, low $\Delta H_{VAP}$ values (i.e., $C^*$-independent $\Delta H_{VAP}$) are suggested to be assumed in the

model to reproduce the measured thermograms.



### 3.3 AMS-derived volatility distribution using PMF

Two organic aerosol groups (SVOA and LVOA) with different volatilities were separated from the AMS data using the PMF method (Sec. 2.3.2). This common two-factor separation is driven by the relative fractions of $m/z$ 44 ($f44$) and $m/z$ 43 ($f43$), connected to the oxidation state (e.g. Aiken et al., 2008). Higher factor solutions associated with other organic groups, commonly determined by PMF analysis, such as biomass burning organic aerosol or hydrocarbon-like organic aerosol, were not pursued. Since this study focuses on volatility distribution of organics using a complex kinetic model and the volatility properties of source dependent components, such as BBOA and HOA, are poorly characterized, we chose to limit the PMF OA components to the main ones clearly connected with oxidation state.

The mass spectra of the two organic groups are shown in Fig. 4. The LVOA mass spectrum shows a highly abundant $m/z$ 44 signal, which mostly corresponds to the $CO_2^+$ ion (Aiken et al., 2008). The mass fraction of $m/z$ 44 shows a good correlation with the O:C ratio in the organic aerosols (Aiken et al., 2008). The SVOA mass spectrum has a high signal at $m/z$ 43, corresponding to $C_2H_3O^+$ ion, which is often considered as a proxy for less oxidized organic aerosol. Hence, the relative abundances of ions at $m/z$ 43 ($f43$) and $m/z$ 44 ($f44$) are our main indicators to separate these two organic groups with different volatilities arising from their different degrees of oxygenation.

Paciga et al. (2016) studied the volatility distribution of an LVOA factor determined by the PMF analysis, and found that a significant amount of the LVOA mass was attributable to ELVOCs with effective saturation concentrations ≤ $10^{-3}$ µg/m$^3$. Hence, probably further advances in the PMF analysis would be needed to assign more than two groups of OA. We tested a three-factor application of PMF, based on the ratio of masses of ions between $m/z$ 44 and $m/z$ 43, and compared the resulting three organics factors with the mass fractions of different organics from the VTDMA data. There was no correlation (R=0.02) between the mass fraction of LVOA from the model and any of PMF three factors. We are not confident to explain, the reason behind this, but it seems possible that the mass spectral statistics based on the PMF classification does not match with the actual volatility grouping. The following discussion thus only focuses on the well-established two-factor PMF solution (SVOA, LVOA) for the organic components.

### 3.4 Comparison between organic aerosol volatility from VTDMA and PMF analysis

#### 3.4.1 General results

In Fig. 5, we compare the organic volatility distributions obtained from the VTDMA data using constant $\Delta H_{VAP}$ values (Combination 1 to 3 in Table 2) with PMF analysis results. Since we used PMF-derived 2-factor results, we summed up the mass fractions of LVOA and ELVOA from the VTDMA for the comparison. The correlation coefficients for the two data sets were relatively similar with





$\Delta H_{VAP}$ values of 60 kJ/mol (R=0.48) and 80 kJ/mol assumed for all organic
groups (R=0.41). Using $\Delta H_{VAP}$ of 100 kJ/mol for all organic groups leads to a
clearly worse correlation (R=0.25) and the model interpreted that the particles
were solely consisting of low volatility organics besides the inorganic species.
Using the enthalpy value of 60 kJ/mol for all organic groups, the modeled mass
fraction of SVOA is higher than the SVOA from the PMF analysis and conversely
lower for LVOA, while using enthalpy value of 80 kJ/mol for organics, the
VTDMA-based OA composition was approximately equal to the ones from the
PMF results. The comparison results differed significantly from the 1:1 line when
the model used $\Delta H_{VAP}$ values of 100 kJ/mol for all organic groups. Using $\Delta H_{VAP}$ of
80 kJ/mol for all three organic groups thus provided the best agreement with the
PMF results. On the other hand, Paciga et al. (2016) studied the volatility
distribution of the PMF-derived organics and estimated that almost half of the
SVOC, which was determined from PMF, is semi-volatile, while 42% is low-
volatile and 6% is extremely low-volatile. This suggests that the two PMF-
derived organic groups, commonly labeled for their oxidation levels, might not
be directly linked to their actual volatilities.

The agreement between the VTDMA- and PMF-based OA volatility distributions
depends on the inorganic mass fractions. The agreement tended to be somewhat
better when the inorganic mass fraction was lower (see Fig. S1). Interestingly,
when the inorganic mass fraction was lower than 0.3, the modeled results
correlated well with the PMF results, with $\Delta H_{VAP}$ values of 100 kJ/mol used in the
model. Results of Häkkinen et al. (2014) suggested that relatively more particle
phase processing, i.e. condensed phase reactions, take place within organic-
inorganic aerosol mixtures having a higher aerosol inorganic mass fraction –
which could be consistent with our results as well.

The use of varying $\Delta H_{VAP}$ values for ELVOA, LVOA and SVOA did not improve the
correlation with the PMF results (see Figs. 6 and S2). Specifically, using $\Delta H_{VAP}$
values from Eq. 3 would result in particles exclusively consisting of low-volatility
organics besides the inorganic species. Lee et al. (2010) reached a similar
conclusion. A single effective $\Delta H_{VAP}$ value can thus well represent the OA mixture.
Cappa and Wilson (2011) studied the volatility of secondary organic aerosol
from the oxidation of α-pinene and reached a similar conclusion: α-pinene SOA
behaved as if it was comprised of a single "meta-compound".

As discussed in Sect. 3.1 we would expect the $C^*$-dependent $\Delta H_{VAP}$ to be the
physically most correct of the alternatives tested – at least when it comes to
simple reversible evaporation. However, if there are other processes in addition
to evaporation taking place in the particle phase upon heating, such as the
molecular decomposition or dissociation of unstable functional groups, the
model might not be able to capture the measured thermogram using Eq. 3. In this
case we might end up with an overestimate in the mass fraction of extremely
low-volatility organics. Donahue et al. (2006) and Riipinen et al. (2010) also
discussed that the evaporation of a mixture is best approximated with
considerably lower effective vaporization enthalpy than the one of a pure
component aerosol. For VTDMA measurements of ambient aerosols with various
compositions and external conditions, the relation between the $C^*$ and



vaporization enthalpy values might be non-linear, species- and/or system-dependent. Moreover, Saleh et al. (2013) reported that the evaporation of particles in laboratory experiments could be simulated using a mass accommodation coefficient much less than one. Hence, also non-unity mass accommodation coefficients of a mixture can add uncertainties to the interpretation of the TD data.

Finally, we compared the median volatility distributions of the organics during the whole campaign using the two methods (Fig. 7). A constant $\Delta H_{VAP}$ value of 80 kJ/mol for all organics was chosen here as the kinetic model input. According to the PMF results, the SVOA contribution to the total organic aerosol mass was around 30%, which is somewhat lower than the SVOA contribution (approximately 40%) obtained based on the VTDMA results. The model estimated that the mass fractions of LVOA and ELVOA of the total OA mass were 34% and 26%, respectively.

### 3.4.2 Time-dependent case studies

Figures 8 and 9 show two case studies for 21 April and 1 May 2014. Time series of mass fractions of the particle constituents from HR-AMS, organic mass fractions from the VTDMA (using Combination 1-3 in Table 2) and PMF analysis are shown.

When the ambient aerosol was dominated by organics (Fig. 8), the modeled SVOA mass fraction followed the temporal pattern of the one determined from PMF analysis. The elevated SVOA mass fraction in the early morning is probably due to the condensation of SVOC onto the particles when temperature was still low, and the following decrease in SVOA after the early morning could be caused by the evaporation of SVOA after the ambient temperature increased. The model-interpreted SVOA mass fraction using $\Delta H_{VAP}$ values of 80 kJ/mol seemed to have somewhat time-delayed effect compared with the one from the PMF analysis.

When the inorganic species dominated the ambient aerosol mass (Fig. 9), a clear diurnal pattern could also be seen from for both the VTDMA and the PMF-derived SVOA and LVOA mass fractions. However, the VTDMA-based mass fraction followed the PMF-derived ones better when using $\Delta H_{VAP}$ values of 60 and 80 kJ/mol compared the one using $\Delta H_{VAP}$ values of 100 kJ/mol (see also Fig. 5). The relative amount of inorganic species in the particle phase might thus affect the particle phase processing. Conclusively, from these two case studies, an effective $\Delta H_{VAP}$ value of 60-80 kJ/mol should be assumed in the model when comparing with the PMF results.

### 4 Summary and conclusions

The volatility of ambient aerosol particles was studied with a Volatility Tandem Differential Mobility Analyzer (VTDMA) in a boreal forest environment in Hyytiälä from April to May of 2014. A kinetic evaporation model was used to further interpret the results and quantify the mass fraction of organics with different volatilities.





When testing the performance of the model against the experimental volatility data, the model was observed to be sensitive to the vaporization enthalpy values of the organics. $C^*$-dependent vaporization enthalpies based on a semi-empirical formula by Epstein et al., 2010 were applied, but the modeled thermograms failed to reproduce the measurements in this case.

The SVOA and LVOA mass fractions obtained from the VTDMA-model results and from the PMF analysis were correlated with each other, and the correlations were best when constant vaporization enthalpies of 60-80 kJ/mol were assumed.

The use of a considerably lower enthalpy value (80kJ/mol) than the semi-empirical ones, the model can best approximate the VTDMA data and the PMF results. Potential explanations to why artificially low vaporization enthalpy values provide the best approximation include thermal decomposition process in addition to evaporation in the particle phase, mixture effects and different mass accommodation coefficients for aerosol mixtures rather than for a pure component system (Riipinen et al., 2010). The interpretation of the VTDMA data using the kinetic evaporation model cannot provide an accurate, definitive volatility distribution for boreal forest aerosols due to the uncertainties in $\Delta H_{VAP}$ and other potential issues mentioned above. However, using a proper effective $\Delta H_{VAP}$ value for OA, the VTDMA-model results nevertheless, for the first time, provide a rough estimate of the volatility for boreal forest aerosols, approximating that around 26% of the monodisperse (100 nm) OA mass is extremely low volatile.

ACKNOWLEDGEMENTS

This work was supported by the Academy of Finland Center of Excellence (grant no. 272041), European Research Council (ATM-NUCLE and ATMOGAIN no. 278277), University of Helsinki funds, and European Commission (ACTRIS, N° 262254).

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





Table 1: Properties of six particle components used as input for the evaporation
model.

| Model input parameter | ELVOA | LVOA | SVOA | Ammonium nitrate (AN) | Ammonium sulfate (AS) | Elemental carbon (EC) |
|---|---|---|---|---|---|---|
| Molar mass, $M_W$ (g/mol) | 300 | 200 | 150 | 80 | 132 | 280 |
| Density, $\rho$ (kg/m³) | 1900 | 1700 | 1400 | 1720 | 1770 | 1900 |
| Surface tension, $\sigma$ (N/m) | 0.05 | 0.05 | 0.05 | 0.05 | 0.05 | 0.05 |
| Diffusion coefficient, $D$ ($10^{-6}$ m²/s) | 5 | 5 | 5 | 5 | 5 | 5 |
| Temperature-dependent factor for $D$, $\mu$ | 1.75 | 1.75 | 1.75 | 1.75 | 1.75 | 1.75 |
| Saturation vapor concentration, $C^*$ (µg/m³) | 1E-5 | 1E-2 | 10 | 76 | 2.0E-3 | 1E-30 |
| Enthalpy of vaporization, $\Delta H_{VAP}$ (kJ/mol) | _a | _a | _a | 152 | 94 | 100 |
| Mass accommodation coefficient, $\alpha_m$ | 1 | 1 | 1 | 1 | 1 | 1 |
| Activity coefficient, $\gamma$ | 1 | 1 | 1 | 1 | 1 | 1 |
| Particle mass for the monodisperse aerosols, $m_P$ (µg/m³)[b] | 0.1 | | | | | |
| Particle mobility diameter, $D_P$ (nm) | 100 | | | | | |

a: The chosen of enthalpy values of three groups of organics are summarized in
Table 2. b: The particle mass in particle size bin of 90-110 nm from DMPS is used
to represent the particle mass for the monodisperse aerosols (i.e. $D_P$ = 100 nm).



Table 2: The combinations of vaporization enthalpy values used as an input for the evaporation model.

|  | ELVOA | LVOA | SVOA |
|---|---|---|---|
| **Combination 1.** | 60 | 60 | 60 |
| **Combination 2.** | 80 | 80 | 80 |
| **Combination 3.** | 100 | 100 | 100 |
| **Combination 4.** | 100 | 80 | 60 |
| **Combination 5.** | 120 | 100 | 80 |
| **Combination 6.** | 130 | 110 | 80 |
| **Combination 7.** | 160 | 130 | 80 |
| **Combination 8.** | 140 | 125 | 100 |
| **Combination 9.** | Eq. 3 | Eq. 3 | Eq. 3 |











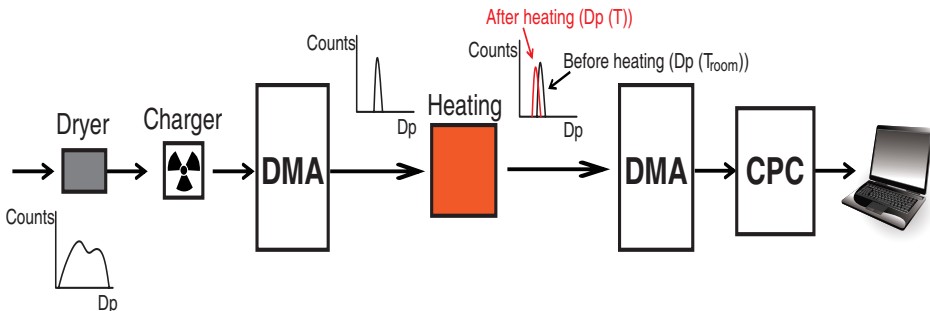

Figure 1: Schematic view of VTDMA system.











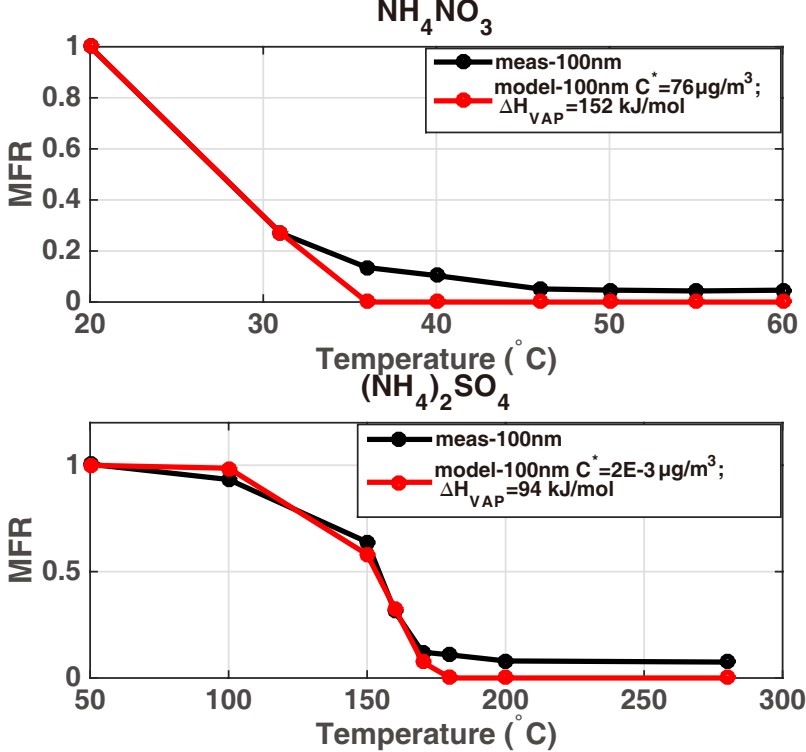

Figure 2: Thermograms of ammonium nitrate and ammonium sulfate using the VTDMA (black lines) and the modeled evaporation using saturation vapor pressures and enthalpies of vaporization corresponding to the best fit with the experimental data (red lines).







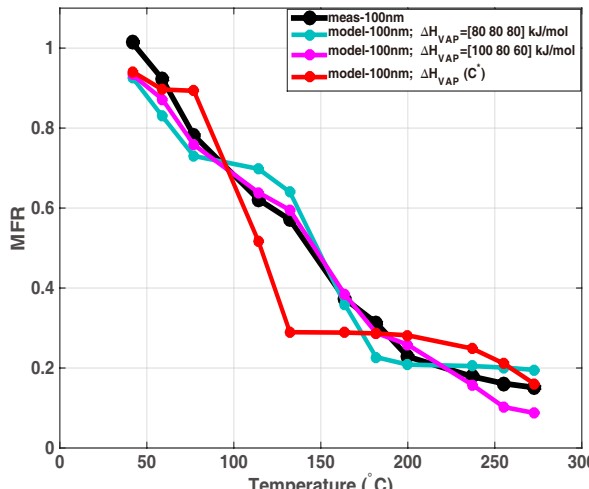

Figure 3: An example of measured (black line) vs. modeled (green, magenta and
red) thermograms assuming different vaporization enthalpies of the organics.






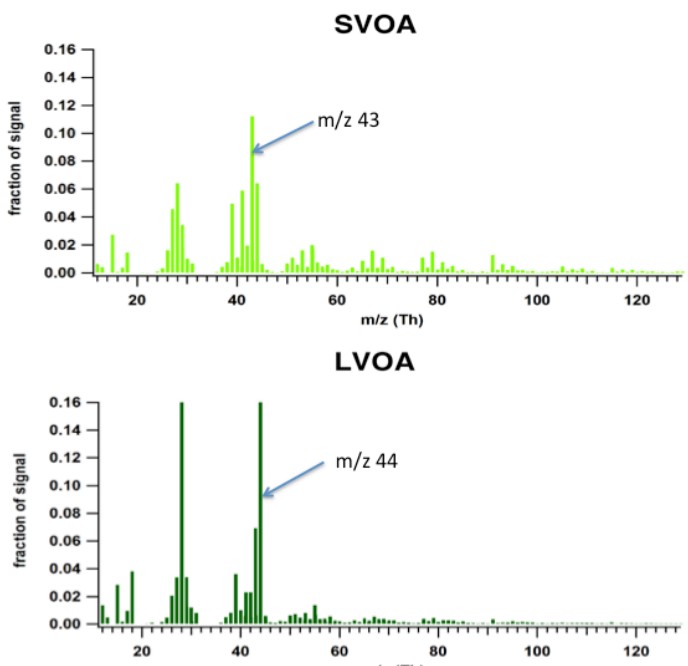

Figure 4: Mass spectrum of SVOA and LVOA obtained from the PMF analysis (two factor solution).



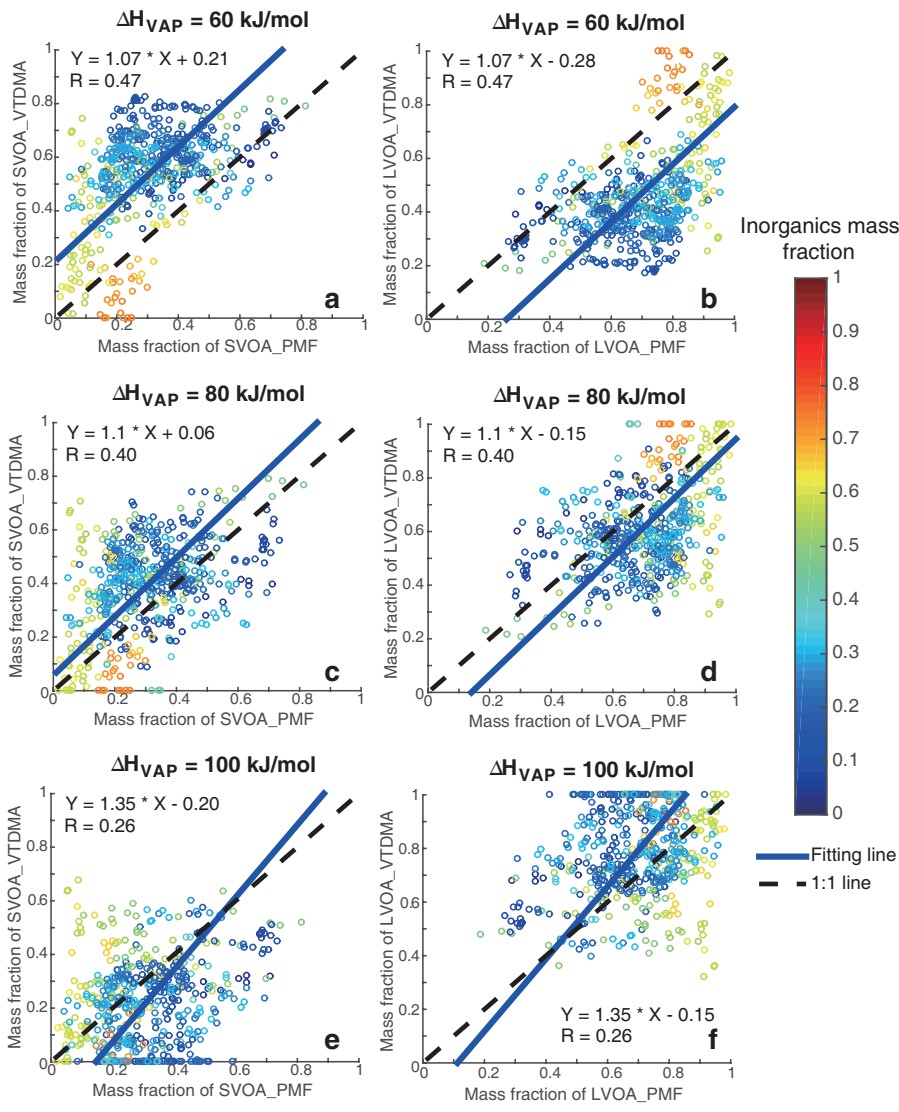


Figure 5: Mass fractions of SVOA and LVOA of the total organic mass obtained from VTDMA data vs. the ones from the PMF analysis. Model results were obtained by using a constant enthalpy value for all organics, corresponding to Combination 1 (a and b), Combination 2 (c and d) and Combination 3 (e and f) in

Table 2. The LVOA_VTDMA here is the sum of LVOA and ELVOA mass fractions. The colors of the data points illustrate the inorganic mass fraction in the particles. Correlation coefficient and equation for the line fitted to the data points are given in the legends.





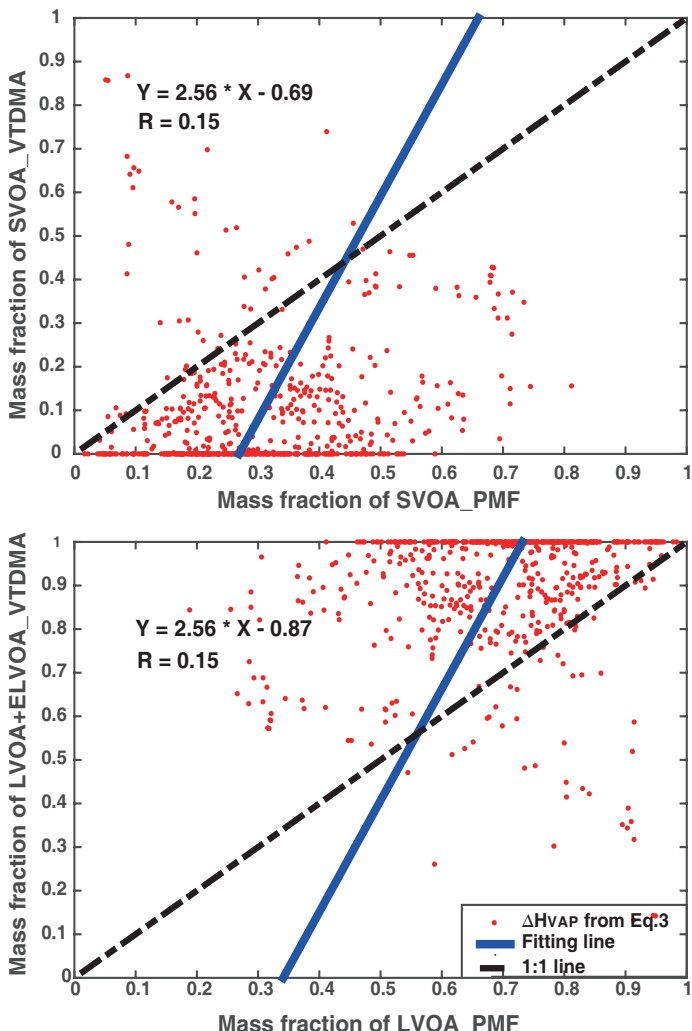


Figure 6: Mass fractions of SVOA and LVOA of the total organic mass obtained from VTDMA data vs. the ones from the PMF analysis. $C^*$-dependent $\Delta H_{vap}$ values based on Eq. 3 were used as the input for the kinetic model. Correlation
coefficient and equation for the line fitted to the data points to describe the agreement between the VTDMA- and PMF-derived organic mass fractions are also given.






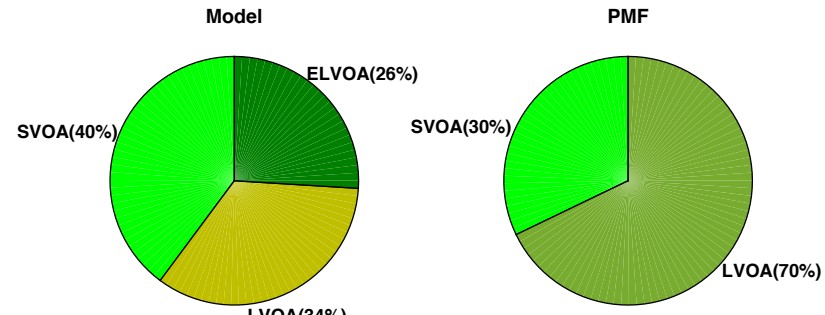


Figure 7: Median organic volatility distribution of the ambient aerosols of this study obtained from the VTDMA data interpreted by the kinetic evaporation model (Riipinen et al., 2010) and the AMS data derived from the PMF analysis.







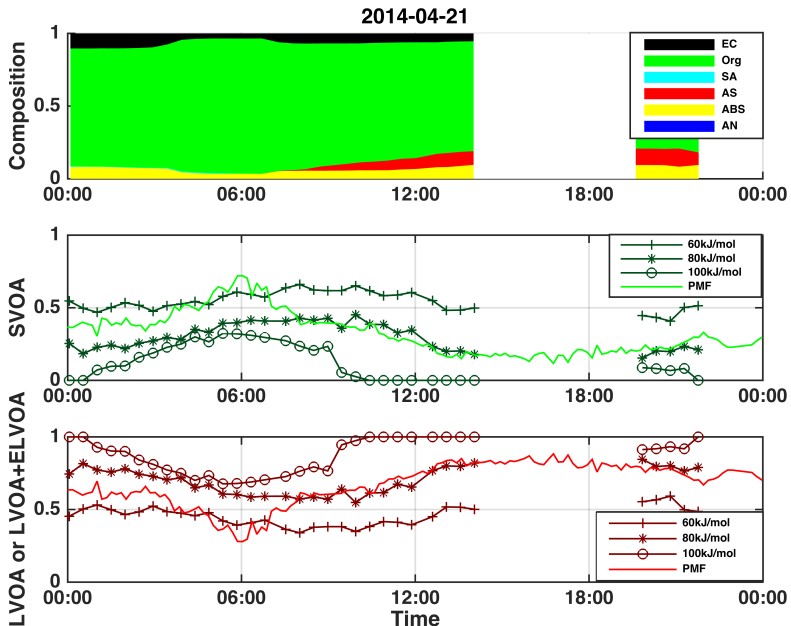


Figure 8: Time series of particle chemical composition obtained from HR-AMS (top), and mass fractions of VTDMA- (the sum of LVOA+ELVOA) and PMF derived SVOA (middle) and LVOA (bottom) on 21 April 2014.





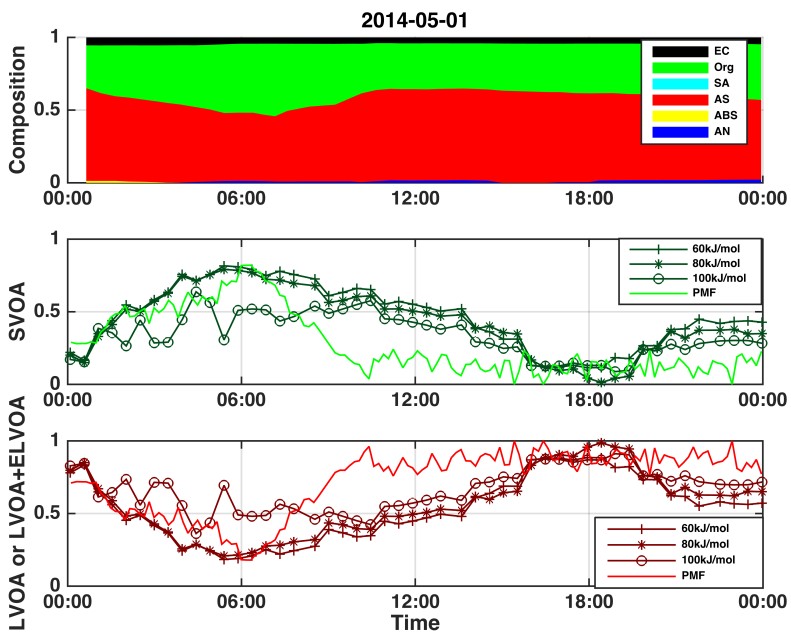

Figure 9: Time series of particle chemical composition obtained from HR-AMS (top), and mass fractions of VTDMA- (the sum of LVOA+ELVOA) and PMF-derived SVOA (middle) and LVOA (bottom) on 01 May 2014.





