# Peer review of "Estimates of the organic aerosol volatility in a boreal forest using two independent methods"

_Atmospheric Chemistry and Physics, 2016_

## Referee Comment (RC1) · Anonymous Referee #2 · 14 Oct 2016

Review of acp-2016-774
Hong et al. "Estimates of organic aerosol volatility in a boreal forest using two independent methods"

**Overview:**
The authors present a nice manuscript utilizing two methods to interpret field measurement data using a kinetic evaporation model and a statistical model. I recommend that their submission be accepted as a paper in *Atmospheric Chemistry and Physics*, after they address the following comments.

**General Comments:**
1) Sections 3.1 and 3.2 appear to be very light on the "discussion" aspect of these "Results and Discussions". What do these results (and their associated figures) provide, beyond a bunch of entries in a table (i.e., what are the implications)?

2) Regarding the authors' identification of the two-factor solution as SVOA and LVOA and the fact that the authors report that the particle concentration was 0.1 $\mu g\ m^{-3}$, it seems to me that it is more likely that the reported two-factor solution from PMF is LVOA and ELVOA. Relating to Specific Comment 15 below, I have serious doubts that a compound having $C^* = 10\ \mu g\ m^{-3}$ can be reliably constrained using the kinetic evaporation model.

3) What are the volatility distributions that were derived? I understand that this is not a generalizable outcome, but perhaps a table summarizing the outcomes from the authors work (including AS, AN, and EC) would clarify this for the reader. This is typically done in the literature, including some of the authors' cited references (e.g., Cappa and Jimenez, 2010; Grieshop et al., 2009; Kuwayama et al., 2015; Lee et al., 2010; May et al., 2013a, 2013b, 2013c; Paciga et al., 2016). Are these the values that the authors are reporting in lines 29-34 in the abstract (this is only obvious in the text of the manuscript in the description of Figure 7)?

**Specific Comments:**
1) Line 95: As written, it almost appears as if the authors are referring to the BBOA factor as a secondary source. Perhaps, a better descriptor for HOA would be "from fossil fuel combustion"?

2) Lines 108-110: In my opinion, the authors should explicitly state that this sensitivity is tested with the kinetic model.

3) Lines 131-134: What are the DMA flow rates, and are the authors concerned with how monodisperse the aerosol population may be given the resulting resolution due to these flows?

4) Line 141: What is the length of the thermodenuder (TD)?

5) Line 141: What kinds of particle losses might be expected in the TD, how significant might they be, and how might these losses, if neglected/uncorrected for, bias the study results?

6) Equation 1: Is the denominator truly at room temperature (25 $^{\circ}$C), or is this really ambient temperature which could fluctuate drastically?

7) Lines 152-154: I recommend that the authors clarify that if VFR = 1 at a given temperature, this implies that they have not evaporated, rather than stating they are non-volatile. Similarly, for VFR = 0, this implies that the particles have fully evaporated at that temperature.

8) Lines 226-227 and 229: The authors have already stated that elemental carbon is abbreviated as "EC" in line 171, so repeating this twice more in these two lines is not necessary.

9) Lines 235-237: I am unfamiliar with the Matlab canned routine *fmincon*, but I am curious if this guarantees a global minimum or if the solver could find local minima instead? For example, due to the uniqueness issue posed by Cappa and Jimenez (2010), May et al. (2013a) utilized a brute-force forward approach to investigate the volatility distribution along with enthalpy of vaporization ($\Delta H_{vap}$) and mass accommodation coefficient ($\alpha$) to determine the global minimum within their solution space using the Riipinen et al. (2010), while Paciga et al. (2016) employ the error minimization approach of Karnezi et al. (2014) to improve the reliability of their solution, also using the Riipinen et al. (2010) model. In my opinion, the authors should comment on their choice of optimization approach and how this could potentially bias their outcomes, if *fmincon* does not guarantee a global minimum in its solution.

10) Lines 244-247: While I understand why the authors are selecting 100 nm as the size to focus on for their analysis, I am curious as to what the overall size distribution of the particles is. Will "arbitrarily" (probably not the right word) selecting a single size bias the outcomes if, for example, the geometric mean diameter of all particles in the sample is 300 nm (since evaporation rates are size dependent)?

11) Lines 266-299: The authors claim that the "volatilities of common inorganic species are relatively well known" in the Introduction (line 69). Therefore, I am wondering what the purpose of going through the process of fitting the saturation concentration (C*) and $\Delta H_{vap}$ is in this work. Is this simply to test the kinetic evaporation model?

12) Lines 319-320: I am curious as to why the authors consider Combinations 4-8 to be "C*-independent" even though $H_{vap}$ is different for each C* for these cases.

13) Lines 335-339: There are a number of studies that characterize the volatility of organic aerosol from individual emission sources, including one by one of the co-authors (May et al., 2013a), so I would argue that this statement is not strictly true as written.

14) Lines 344-346: The authors appear to be implying that $C_3H_7^+$ is negligible at *m/z* 43. Is this true?

15) Table 1: where does the value of "particle total mass" come from? And is this really the total mass concentration (0.1 $\mu$g m$^{-3}$ seems very low)? If so, do the authors have any hope of actually constraining the SVOA component? For C* = 10 $\mu$g m$^{-3}$, the predicted mass fraction in the particle phase is 1%, following Donahue et al. (2006). If truly only 1% of the SVOA mass is in the particle phase, how much certainty do the authors have in their analysis?

16) Figures 2-3: If the initial temperature set point in the TD is 25 ºC (line 144), why are the initial data points 20 ºC, 50 ºC, and ~40 ºC in these figures? This inconsistency is confusing.

17) Figures 2-3: This appears to be a little messy with marker-and-line combinations representing both experimental data and model outputs. I recommend, for example, changing the data to markers and the predictions to lines.

18) Figure 5: First, I would recommend that the authors clarify that the y-axis represents interpretation using the kinetic model and that the x-axis represents interpretation using the statistical model (PMF). Second, something that I find curious is that the slopes of both columns are identical, but the offsets are different. Does this indicate a systematic bias or is this an artifact of there only being two factors in the comparison?

19) Figure 6: The Epstein et al. (2010) C*- $\Delta H_{vap}$ has been trashed relentlessly in studies probing $\Delta H_{vap}$ because it is based on pure components, and the relationship clearly doesn't work for mixtures. I don't think that we as a community need yet another figure demonstrating this (but having some discussion of this in the text is fine).

20) Figure 7: First, both pie charts are derived from models, so I suggest that the labels are changed to say something like "Kinetic model results" for the left and "Statistical model results (PMF)" for the right.

Second, in the caption, it would be useful for the reader if the authors state to which value of $\Delta H_{vap}$ the kinetic model results correspond.

Third, if this is indeed really LVOA and ELVOA that is being identified in PMF (see General Comment #3 above), then the kinetic model outcome is biased by the authors' assumed definitions of the PMF model results. This potential bias should be resolved either explicitly or implicitly as the authors respond to General Comment #3 and Specific Comment #15.

21) Figures 8-9: What are the implications here? Is, for example, an effective $\Delta H_{vap}$ = 80 kJ mol$^{-1}$ the optimum value that is representative of ambient organic aerosol at the sampling site? I'm not really sure how to interpret these figures without some additional context, either in the captions or in the body of the manuscript itself.

**References:**
Cappa, C. D. and Jimenez, J. L.: Quantitative estimates of the volatility of ambient organic aerosol, Atmos. Chem. Phys., 10(12), 5409–5424, doi:10.5194/acp-10-5409-2010, 2010.
Donahue, N. M., Robinson, A. L., Stanier, C. O. and Pandis, S. N.: Coupled Partitioning, Dilution, and Chemical Aging of Semivolatile Organics, Environ. Sci. Technol., 40(8), 2635–2643, doi:10.1021/es052297c, 2006.
Epstein, S. A., Riipinen, I. and Donahue, N. M.: A semiempirical correlation between enthalpy of vaporization and saturation concentration for organic aerosol., Environ. Sci. Technol., 44(2), 743–748, doi:10.1021/es902497z, 2010.
Grieshop, A. P., Miracolo, M. A., Donahue, N. M. and Robinson, A. L.: Constraining the Volatility Distribution and Gas-Particle Partitioning of Combustion Aerosols Using Isothermal Dilution and Thermodenuder Measurements, Environ. Sci. Technol., 43(13), 4750–4756, doi:10.1021/es8032378, 2009.
Karnezi, E., Riipinen, I. and Pandis, S. N.: Measuring the atmospheric organic aerosol volatility distribution: a theoretical analysis, Atmos. Meas. Tech., 7(9), 2953–2965, doi:10.5194/amt-7-2953-2014, 2014.
Kuwayama, T., Collier, S., Forestieri, S., Brady, J. M., Bertram, T. H., Cappa, C. D., Zhang, Q. and Kleeman, M. J.: Volatility of Primary Organic Aerosol Emitted from Light Duty Gasoline Vehicles, Environ. Sci. Technol., 49(3), 1569–1577, doi:10.1021/es504009w, 2015.
Lee, B. H., Kostenidou, E., Hildebrandt, L., Riipinen, I., Engelhart, G. J., Mohr, C., DeCarlo, P. F., Mihalopoulos, N., Prevot, a. S. H., Baltensperger, U. and Pandis, S. N.: Measurement of the ambient organic aerosol volatility distribution: application during the Finokalia Aerosol Measurement Experiment (FAME-2008), Atmos. Chem. Phys., 10(24), 12149–12160, doi:10.5194/acp-10-12149-2010, 2010.
May, A. A., Levin, E. J. T., Hennigan, C. J., Riipinen, I., Lee, T., Collett, J. L., Jimenez, J. L., Kreidenweis, S. M. and Robinson, A. L.: Gas-particle partitioning of primary organic aerosol emissions: 3. Biomass burning, J. Geophys. Res. Atmos., 118(19), 11,327-11,338, doi:10.1002/jgrd.50828, 2013a.
May, A. A., Presto, A. A., Hennigan, C. J., Nguyen, N. T., Gordon, T. D. and Robinson, A. L.: Gas-particle partitioning of primary organic aerosol emissions: (1) Gasoline vehicle exhaust, Atmos. Environ., 77, 128–139, doi:10.1016/j.atmosenv.2013.04.060, 2013b.
May, A. A., Presto, A. A., Hennigan, C. J., Nguyen, N. T., Gordon, T. D. and Robinson, A. L.: Gas-particle partitioning of primary organic aerosol emissions: (2) diesel vehicles., Environ. Sci. Technol., 47(15),

8288–96, doi:10.1021/es400782j, 2013c.

Paciga, A., Karnezi, E., Kostenidou, E., Hildebrandt, L., Psichoudaki, M., Engelhart, G. J., Lee, B.-H., Crippa, M., Prévôt, A. S. H., Baltensperger, U. and Pandis, S. N.: Volatility of organic aerosol and its components in the megacity of Paris, Atmos. Chem. Phys., 16(4), 2013–2023, doi:10.5194/acp-16-2013-2016, 2016.

Riipinen, I., Pierce, J. R., Donahue, N. M. and Pandis, S. N.: Equilibration time scales of organic aerosol inside thermodenuders: Evaporation kinetics versus thermodynamics, Atmos. Environ., 44(5), 597–607, doi:10.1016/j.atmosenv.2009.11.022, 2010.

---

## Referee Comment (RC2) · Anonymous Referee #1 · 22 Dec 2016

General Comments

The authors have estimated volatility distribution of ambient SOA in a boreal forest by interpreting Volatility Tandem Differential Mobility Analyzer (VTDMA) measurements with a kinetic evaporation model. The modeling results show that 40% of the SOA was semi-volatile, 34% low-volatile, and 26% extremely low-volatile, with an effective enthalpy of vaporization value of 80 kJ/mole for all organics. They also independently estimated semi-volatile and low-volatile organic mass fractions by applying Positive Matrix Factorization (PMF) to the High Resolution Aerosol Mass Spectrometer data, with the factor separation based on the oxygenation levels of organics (relative abundance of mass ions at m/z 43 and m/z 44). The manuscript is well written, but the usefulness of the results is questionable. The study could become publishable if the following comments are satisfactorily addressed.

[Figure]

Detailed Comments

1) It should be made clear in the abstract, body, and conclusion that this study estimates the effective volatility distribution of SOA after it has formed and undergone aging (particle-phase reactions) and not the volatility distribution of the condensing organic gases that produced the SOA. While the authors do not imply the latter, some readers may inadvertently misinterpret the results and incorrectly apply them for modeling purposes if the caveat is not explicitly stated.

This then brings up the question of the usefulness of the results. If the estimated volatility distribution of SOA does not represent the volatility distribution of the condensing organics then where and how would one use this information? What is it that we have learned from this exercise that is of value? To drive this point further, it is possible that the semi-volatile fraction of the condensing organics may be more than 40%, such that some then undergoes particle-phase reactions to form low- and extremely low-volatile compounds. Upon heating the SOA in the VTDMA, these newly formed compounds (e.g., dimers, oligomers) may partially decompose back to the original species or may even fragment (especially at higher temperatures) to something completely different before evaporating. However, these processes are not examined in the present study. The evaporation model simply assumes that the three lumped species (with different volatilities) do not chemically interact. Thus, it is difficult to draw any useful or meaningful conclusions out of the present analysis.

2) Furthermore, since the VTDMA experiments were carried out under dry conditions, the boreal forest SOA is expected to be viscous (Virtanen et al. 2010 Nature), especially at 25 C, such that there would be significant particle-phase diffusion limitation for the evaporating species. Perhaps the evidence for this limitation is present in Figure 3, which shows that the model tends to be way more evaporative than observed at the lower temperatures–in fact, the first data point indicates no evaporation (MFR = 1). This can potentially skew the effective volatility distribution estimated by the model quite a bit.

3) It is stated that the residence time inside the thermodenuder was around 2.5 s. Does this mean that the evaporation model was run for just 2.5 s to simulate each data point? Also, does the model assume that the aerosol instantly reaches the targeted temperature the moment it enters the thermodenuder? Can the authors estimate the time it takes the aerosol to reach the target temperature? The model should account for it if it's comparable to the residence time.

4) Please show error bars on the VTDMA measurements displayed in Figures 2 and 3.

5) The authors state that the agreement between the VTDMA results and the PMF-derived results as reasonable when it is quite the opposite. The linear correlation coefficient (r) of 0.4 indicates there is not a good correlation between the VTDMA- and PMF-based results (Figure 5c,d). The coefficient of determination (rˆ2) is only 0.16, which means only 16% of the variation can be explained by the linear relationship between the two methods. It is clear that the comparison of VTDMA and PMF results was not quite successful. I suggest that the authors revise the text at all the appropriate places and describe the results of correlation as they are.

---

## Author Comment (AC2) · 6 Feb 2017

Answers to Referee #2

The authors appreciate the time the reviewer has spent on our manuscript, helping us produce a higher quality, understandable publication. All the requested corrections and suggestions are addressed and introduced to the revised version of the manuscript.

Detailed Comments

1) It should be made clear in the abstract, body, and conclusion that this study estimates the effective volatility distribution of SOA after it has formed and undergone aging (particle-phase reactions) and not the volatility distribution of the condensing organic gases that produced the SOA. While the authors do not imply the latter, some readers may inadvertently misinterpret the results and incorrectly apply them for modeling purposes if the caveat is not explicitly stated.

Reply: This is correct, and this issue will be clarified through the whole manuscript.

This then brings up the question of the usefulness of the results. If the estimated volatility distribution of SOA does not represent the volatility distribution of the condensing organics then where and how would one use this information? What is it that we have learned from this exercise that is of value? To drive this point further, it is possible that the semi-volatile fraction of the condensing organics may be more than 40%, such that some then undergoes particle-phase reactions to form low- and extremely low-volatile compounds. Upon heating the SOA in the VTDMA, these newly formed compounds (e.g., dimers, oligomers) may partially decompose back to the original species or may even fragment (especially at higher temperatures) to something completely different before evaporating. However, these processes are not examined in the present study. The evaporation model simply assumes that the three lumped species (with different volatilities) do not chemically interact. Thus, it is difficult to draw any useful or meaningful conclusions out of the present analysis.

Reply: Indeed, the volatility distribution of SOA studied here did not represent the volatility distribution of the condensing organic compounds in the gaseous phase. However, it provides "one side of the story" in the form of insights into the volatility (and hence e.g. the evaporation potential) of the compounds that are present in the particle phase (of course with the caveat of the effects of the elevated temperature). This will be useful input for closure studies combining this information with

condensation studies aiming to derive how the aerosol size distributions are affected by given gas-phase species. In fact, we are working on such a closure study for future. Furthermore, the results are useful to compare the volatility of the boreal forest aerosol to similar results from other sites (e.g. Cappa and Jimenez 2010; Cappa and Wilson, 2011; May et al., 2013a). Finally, it also provides insight into the usefulness and applicability (and limitations) of TD setups for inferring information about aerosol volatility. We will add discussion on these issues to the revised manuscript.

2) Furthermore, since the VTDMA experiments were carried out under dry conditions, the boreal forest SOA is expected to be viscous (Virtanen et al. 2010 Nature), especially at 25 C, such that there would be significant particle-phase diffusion limitation for the evaporating species. Perhaps the evidence for this limitation is present in Figure 3, which shows that the model tends to be way more evaporative than observed at the lower temperatures–in fact, the first data point indicates no evaporation (MFR = 1). This can potentially skew the effective volatility distribution estimated by the model quite a bit.

Reply: The referee is right that the kinetics of evaporation of non-liquid particles may be somewhat affected by the diffusion coefficient of a viscous solution (Tong et al., 2011). However, it should be noted that the ambient data were mixtures of organics, water and inorganics, for which it is difficult to quantify the potential diffusivity impacts. We will add brief discussion on this issue to the revised manuscript.

3) It is stated that the residence time inside the thermodenuder was around 2.5 s. Does this mean that the evaporation model was run for just 2.5 s to simulate each data point? Also, does the model assume that the aerosol instantly reaches the targeted temperature the moment it enters the thermodenuder? Can the authors estimate the time it takes the aerosol to reach the target temperature? The model should account for it if it's comparable to the residence time.

Reply: Yes, the evaporation model was run for 2.5s to simulate each data point, since measured evaporation time is 2.5 s in this case. The temperature profile was measured with a temperature sensor (Rotronic HC2-CO4) inside the heating tube at flow rate of 2 L/min with setting temperature at 100 °C (see Fig.1).  As shown in Fig. 1, the temperature did not reached the targeted temperature at the moment the aerosols entered the thermodenuder but increased slowly at the entrance of the heating tube and reached the targeted temperature ±5 °C at around 20 cm

from the entrance. The temperature stayed at this value before falling near the exit, 45 cm from the entrance of the heating tube. This distance with temperature within ±5 °C of the targeted one was used for calculation the residence time of the heating section. The obtained residence time was then applied in the model analysis. It was assumed that the particles were instantaneously thermally equilibrated with the surrounding gas phase. We think this is a reasonable assumption with respect to the time scales relevant for the evaporation, as the system was in atmospheric pressure

4) Please show error bars on the VTDMA measurements displayed in Figures 2 and 3.

Reply: We will add error bars on the VTDMA measurements data to the revised manuscript.

5) The authors state that the agreement between the VTDMA results and the PMF- derived results as reasonable when it is quite the opposite. The linear correlation coefficient (r) of 0.4 indicates there is not a good correlation between the VTDMA- and PMF-based results (Figure 5c,d). The coefficient of determination (r^2) is only 0.16, which means only 16% of the variation can be explained by the linear relationship between the two methods. It is clear that the comparison of VTDMA and PMF results was not quite successful. I suggest that the authors revise the text at all the appropriate places and describe the results of correlation as they are.

Reply: Thank you for pointing this out, we agree. Consequently, we will make revisions in line 44-47 of the revised manuscript:

'In general, the best agreement between the VTDMA results and the PMF-derived mass fractions of organics was obtained when $\Delta H_{VAP} = 80$ kJ/mol was set for all organic groups in the model, with a linear correlation coefficient of around 0.4. However, this still indicates that only about 16% ($R^2$) of the variation can be explained by the linear regression between the results from these two methods.'

Discussion in line 388-400 will be revised as: 'Using the enthalpy value of 60 kJ/mol for all organic groups, the modeled mass fraction of SVOA was higher than the SVOA from the PMF analysis. The opposite is true for LVOA, while using $\Delta H_{VAP}$ values of 100 kJ/mol for all organic groups, the comparison results differed significantly from the 1:1 line. With enthalpy value of 80 kJ/mol for organics, the VTDMA-based OA

composition was approximately equal to the ones from the PMF results, however, with a linear correlation coefficient of only 0.4. This relatively low correlation coefficient suggests that additional information on each of the method is needed to analyze the potential links between the AMS and volatility data. Moreover, Paciga et al. (2016) studied the volatility distribution of the PMF-derived organics and estimated that almost half of the SVOC, which was determined from PMF, is semi-volatile, while 42% is low-volatile and 6% is extremely low-volatile. This suggests that the two PMF-derived organic groups, commonly labeled for their oxidation levels, might not be directly linked to their actual volatilities.

[Figure]

Fig.1. Temperature profile along the axis of the heating section at a flow rate of 2 L/min.

Reference:

Cappa, C. D., and Jimenez, J. L.: Quantitative estimate of the volatility of ambient organic aerosol, Atmos. Chem. Phys., 10, 5409-5424, doi:10.5194/acp-10-5409-2010, 2010.

Cappa, C. D., and Wilson, K. R.: Evolution of organic aerosol mass spectra upon heating: implications for OA phase and partitioning

behavior, Atmos. Chem. Phys., 11, 1895-1911, doi:10.5194/acp-11-1895-2011, 2011.

May, A. A., Levin, E. J. T., Hennigan, C. J., Riipinen, I., Lee, T., Collett, J. L., Jimenez, J. L., Kreidenweis, S. M. and Robinson, A. L.: Gas-particle partitioning of primary organic aerosol emissions: 3. Biomass burning, J. Geophys. Res. Atmos., 118(19), 11,327-11,338, doi:10.1002/jgrd.50828, 2013a.

Paciga, A., Karnezi, E., Kostenidou, E., Hildebrandt, L., Psichoudaki, M., Engelhart, G. J., Lee, B.-H., Crippa, M., Prevot, A. S. H., Baltensperger, U., and Pandis, S. N.: Volatility of organic aerosol and its components in the megacity of Paris, Atmos, Chem. Phys., 16, 2013-2023, doi:10.5194/acp-16-2013-2016, 2016.

Tong, H.-J., Reid, J. P., Bones, D. L., Luo, B. P., and Krieger, U. K.: Measurements of the timescales for the mass transfer of water in glassy aerosol at low relative humidity and ambient temperature, Atmos. Chem. Phys., 11, 4739–4754, doi:10.5194/acp-11-4739- 2011, 2011.

---

## Author Response (AR1)

Answers to Referee #1

The authors appreciate the time the reviewer has spent on our manuscript, helping us to produce a higher quality, understandable publication. All the requested corrections and suggestions are addressed and introduced to the revised version of the manuscript.

**General Comments:**

1) Sections 3.1 and 3.2 appear to be very light on the "discussion" aspect of these "Results and Discussions". What do these results (and their associated figures) provide, beyond a bunch of entries in a table (i.e., what are the implications)?

Reply: Even though there have been several earlier studies (Huffman et al., 2009) reporting the evaporation of ammonium sulfate, the specific C* and ΔHvap values are scarce or not available at all. We also wanted to compare results obtained with our technique to previous studies. In Sect. 3.2, the main scope was to evaluate the performance of the model. It is reported that the model is sensitive towards ΔHvap, hence, sensitivity analysis towards this parameter was included in this section. More discussion on the choice of the specific ΔHvap was given in Sect. 3.4.1. We will highlight these aspects in the revised manuscript.

2) Regarding the authors' identification of the two-factor solution as SVOA and LVOA and the fact that the authors report that the particle concentration was 0.1 μg m$^{-3}$, it seems to me that it is more likely that the reported two-factor solution from PMF is LVOA and ELVOA. Relating to Specific Comment 15 below, I have serious doubts that a compound having C* = 10 μg m$^{-3}$ can be reliably constrained using the kinetic evaporation model.

Reply: The information the manuscript gave is perhaps misleading: the average total particle concentration from the dataset used was in fact 2.90 μg m$^{-3}$, and the organic aerosol mass is 1.96 μg m$^{-3}$. The value the manuscript reported was the mass concentration of the selected monodisperse aerosol particles (100 nm in diameter). We will clarify this in the revised manuscript.

3) What are the volatility distributions that were derived? I

understand that this is not a generalizable outcome, but perhaps a table summarizing the outcomes from the authors work (including AS, AN, and EC) would clarify this for the reader. This is typically done in the literature, including some of the authors' cited references (e.g., Cappa and Jimenez, 2010; Grieshop et al., 2009; Kuwayama et al., 2015; Lee et al., 2010; May et al., 2013a, 2013b, 2013c; Paciga et al., 2016). Are these the values that the authors are reporting in lines 29-34 in the abstract (this is only obvious in the text of the manuscript in the description of Figure 7)?

Reply: Yes, the values we gave in Fig. 7 in the manuscript were the volatility distributions obtained. The values are the median values of the dataset taken during the whole campaign.

**Specific Comments:**

1) Line 95: As written, it almost appears as if the authors are referring to the BBOA factor as a secondary source. Perhaps, a better descriptor for HOA would be "from fossil fuel combustion"?

Reply: We agree that the statement here was not written in a clear way. The sentence will be changed as: 'Typical organic groups determined using the PMF analysis include e.g. hydrocarbon-like OA (HOA), biomass burning OA (BBOA) and cooking OA (COA) or oxygenated OA (OOA).'

2) Lines 108-110: In my opinion, the authors should explicitly state that this sensitivity is tested with the kinetic model.

Reply: We will change this part into: 'The sensitivity of the kinetic model was tested towards different parameters of organic compounds, including density, molar mass, saturation vapor concentration, and diffusion coefficient.

3) Lines 131-134: What are the DMA flow rates, and are the authors concerned with how monodisperse the aerosol population may be given the resulting resolution due to these flows?

Reply: The aerosol flow rate of DMAs in our system was 1 l/min, while the sheath flow of the DMAs was kept at 10 l/min. Such flow configuration is quite commonly utilized in the TDMA community. Applying the Stolzenburg kernels with the selected dry sizes and

these flow rates into the DMA and assuming that full width of the peak width at half maximum (FWHM) describes well the width of the transfer function, the following width was obtained:100 nm ±2.9 nm. Therefore the particles were monodisperse within ±3 %. However, this spread was already taken into account in the inversion toolkit by Gysel et al. (2009) in the data analysis.

4) Line 141: What is the length of the thermodenuder (TD)?

Reply: The total length of the thermodenuder (TD) is 50 cm.

5) Line 141: What kinds of particle losses might be expected in the TD, how significant might they be, and how might these losses, if neglected/uncorrected for, bias the study results?

Reply: The major loss processes in the heating tube are caused by thermophoresis and Brownian diffusion. According to Ehn et al. (2007), who used a similar TD as ours, the losses for aerosol particles above 15 nm in diameter were observed to be less than 20% when heated to 280 °C. Due to these losses, we might indeed underestimate the mass concentration of the monodisperse aerosol particles after heating. However, our study was focusing on the change in particle size, which should not be affected very much by the losses. We will add a brief discussion of the losses to the revised manuscript.

6) Equation 1: Is the denominator truly at room temperature (25 °C), or is this really ambient temperature, which could fluctuate drastically?

Reply: Yes, the size of the original particles was selected by the first DMA at room temperature. The room temperature, where the first DMA located was set and maintained at 25 °C ±2 °C.

7) Lines 152-154: I recommend that the authors clarify that if VFR = 1 at a given temperature, this implies that they have not evaporated, rather than stating they are non-volatile. Similarly, for VFR = 0, this implies that the particles have fully evaporated at that temperature.

Reply: We agree. The statement will be corrected as: 'With VFR = 1 at a given temperature, we consider particles have not evaporated, while with VFR = 0 particles are considered to fully evaporate upon

heating at that temperature.'

8) Lines 226-227 and 229: The authors have already stated that elemental carbon is abbreviated as "EC" in line 171, so repeating this twice more in these two lines is not necessary.

Reply: The abbreviation 'EC' will be used here instead.

9) Lines 235-237: I am unfamiliar with the Matlab canned routine *fmincon*, but I am curious if this guarantees a global minimum or if the solver could find local minima instead? For example, due to the uniqueness issue posed by Cappa and Jimenez (2010), May et al. (2013a) utilized a brute-force forward approach to investigate the volatility distribution along with enthalpy of vaporization ($\Delta H_{vap}$) and mass accommodation coefficient ($\alpha$) to determine the global minimum within their solution space using the Riipinen et al. (2010), while Paciga et al. (2016) employ the error minimization approach of Karnezi et al. (2014) to improve the reliability of their solution, also using the Riipinen et al. (2010) model. In my opinion, the authors should comment on their choice of optimization approach and how this could potentially bias their outcomes, if *fmincon* does not guarantee a global minimum in its solution.

Reply: The fmincon function indeed does not guarantee a global minimum. However, this was tested for by changing the initial guesses the function was run with and it was found that the solution we got was dependent on the initial guess we used. To guarantee the uniqueness of the fit, we used only three volatility bins in the fits. Furthermore, the optimization method was constrained with the mass fraction of each organic group and the total measured mass fraction of organics from AMS data. With those constrains, fmincon finds the best solution the computer can give and will be quite close to the global minimum.

10) Lines 244-247: While I understand why the authors are selecting 100 nm as the size to focus on for their analysis, I am curious as to what the overall size distribution of the particles is. Will "arbitrarily" (probably not the right word) selecting a single size bias the outcomes if, for example, the geometric mean diameter of all particles in the sample is 300 nm (since evaporation rates are size dependent)?

Reply: The average geometric mean dry diameter of the overall size distribution of boreal forest aerosols was 60-200 nm if two-mode fit was applied to the measured number size distribution data (Asmi et al., 2011). We therefore expect that the 100 nm particles were relatively representative of the typical size distributions at the studied site. According to Hong et al. (2014), we observed a size-dependent evaporation between the nucleation mode and accumulation mode particles using similar VTDMA setup, however, size-dependent chemical composition information of aerosol mass is also needed to give conclusive statement regarding to their volatility distribution.

11) Lines 266-299: The authors claim that the "volatilities of common inorganic species are relatively well known" in the Introduction (line 69). Therefore, I am wondering what the purpose of going through the process of fitting the saturation concentration (C*) and $\Delta H_{vap}$ is in this work. Is this simply to test the kinetic evaporation model?

Reply: As specified in General comment 1, the C* and $\Delta H_{vap}$ values were inferred to evaluate our approach. These parameters were also used for the model input to simulate the evaporation of ambient aerosols.

12) Lines 319-320: I am curious as to why the authors consider Combinations 4-8 to be "C*-independent" even though $H_{vap}$ is different for each C* for these cases.

Reply: The reviewer is correct. We will revise the wording in the manuscript accordingly.

13) Lines 335-339: There are a number of studies that characterize the volatility of organic aerosol from individual emission sources, including one by one of the co-authors (May et al., 2013a), so I would argue that this statement is not strictly true as written.

Reply: The statement here was indeed not clear and we will remove it from the revised manuscript.

14) Lines 344-346: The authors appear to be implying that $C_3H_7^+$ is negligible at $m/z$ 43. Is this true?

Reply: Ng et al. (2011) stated that "The m/z 43 fragment is mainly $C_2H_3O^+$ for the OOA component, and $C_3H_7^+$ for the HOA component." and according to Crippa et al. (2014), the HOA contribution in Hyytiälä is low (6-7%) compared to the oxidized species with significant m/z 43 contribution, SV-OOA (34-37%). Hence, we believe $C_2H_3O^+$ is the dominant ion at m/z 43 over $C_3H_7+$. Moreover, as both of the ions are indicative of low oxidation level species (Ng et al., 2011), the exact molecular composition of m/z 43 "tracer" signal does not matter either.

15) Table 1: where does the value of "particle total mass" come from? And is this really the total mass concentration (0.1 µg m$^{-3}$ seems very low)? If so, do the authors have any hope of actually constraining the SVOA component? For C* = 10 µg m$^{-3}$, the predicted mass fraction in the particle phase is 1%, following Donahue et al. (2006). If truly only 1% of the SVOA mass is in the particle phase, how much certainty do the authors have in their analysis?

Reply: See the answer to General comments #2. The value of 0.1 µg is the mass concentration of the monodisperse aerosol particles (100 nm in diameter), which was calculated from DMPS data. This was done by integrating the particle number size concentration within 90-110 nm multiplying a constant particle density of 1.2 kg/m$^3$, and represented this value as the monodisperse aerosol mass concentration.

16) Figures 2-3: If the initial temperature set point in the TD is 25 ºC (line 144), why are the initial data points 20 ºC, 50 ºC, and ~40 ºC in these figures? This inconsistency is confusing.

Reply: For ambient measurements, the aerosols were brought to a room at 25 $^O$C. For AN and AS, the evaporation measurements were performed in laboratory conditions, where lower temperatures can be achieved, since AN might already evaporate below 25 $^O$C. We will modify the figure and its caption in the revised manuscript to avoid the confusion.

17) Figures 2-3: This appears to be a little messy with marker-and-line combinations representing both experimental data and model outputs. I recommend, for example, changing the data to markers

and the predictions to lines.

Reply: We will change the figure as suggested in the revised manuscript.

18) Figure 5: First, I would recommend that the authors clarify that the y-axis represents interpretation using the kinetic model and that the x-axis represents interpretation using the statistical model (PMF). Second, something that I find curious is that the slopes of both columns are identical, but the offsets are different. Does this indicate a systematic bias or is this an artifact of there only being two factors in the comparison?

Reply: We will clarify the axes more clearly in the manuscript. From line 827, we will add the following statement: 'Here, the Y-axis represents the VTDMA results interpretation using the kinetic model and the X-axis represents the AMS results interpretation using the statistical model (PMF)'. The different intercepts are more likely related to the fact that there were only two volatility classes that the particles were assumed to consist of.

19) Figure 6: The Epstein et al. (2010) C*- $\Delta H_{vap}$ has been trashed relentlessly in studies probing $\Delta H_{vap}$ because it is based on pure components, and the relationship clearly doesn't work for mixtures. I don't think that we as a community need yet another figure demonstrating this (but having some discussion of this in the text is fine).

Reply: We agree and will move Fig.6 to the supplement.

20) Figure 7: First, both pie charts are derived from models, so I suggest that the labels are changed to say something like "Kinetic model results" for the left and "Statistical model results (PMF)" for the right.

Reply: The legend of Fig. 7 in the manuscript will be changed as: 'Kinetic model results' on the left and 'Statistical model results (PMF)' on the right.

Second, in the caption, it would be useful for the reader if the authors state to which value of $\Delta H_{vap}$ the kinetic model results correspond.

Reply: We will add '$\Delta H_{vap}$ = 80 kJ/mol was used in the kinetic evaporation model' in the figure caption.

Third, if this is indeed really LVOA and ELVOA that is being identified in PMF (see General Comment #3 above), then the kinetic model outcome is biased by the authors' assumed definitions of the PMF model results. This potential bias should be resolved either explicitly or implicitly as the authors respond to General Comment #3 and Specific Comment #15.

Reply: This issue was answered both in General comment #3 and specific Comment #15.

21) Figures 8-9: What are the implications here? Is, for example, an effective $\Delta H_{vap}$ = 80 kJ mol$^{-1}$ the optimum value that is representative of ambient organic aerosol at the sampling site? I'm not really sure how to interpret these figures without some additional context, either in the captions or in the body of the manuscript itself.

Reply: We will modify line 471 as 'These two case studies suggest that an effective $\Delta H_{VAP}$ value of 60-80 kJ/mol represent the boreal forest organic aerosols best.'

'In general, the best agreement between the VTDMA results and the PMF-derived mass fractions of organics was obtained when $\Delta H_{VAP} = 80$ kJ/mol was set for all organic groups in the model, with a linear correlation coefficient of around 0.4. However, this still indicates that only about 16% ($R^2$) of the variation can be explained by the linear regression between the results from these two methods.'

Discussion in line 388-400 will be revised as: 'Using the enthalpy value of 60 kJ/mol for all organic groups, the modeled mass fraction of SVOA was higher than the SVOA from the PMF analysis. The opposite is true for LVOA, while using $\Delta H_{VAP}$ values of 100 kJ/mol for all organic groups, the comparison results differed significantly from the 1:1 line. With enthalpy value of 80 kJ/mol for organics, the VTDMA-based OA

composition was approximately equal to the ones from the PMF results, however, with a linear correlation coefficient of only 0.4. This relatively low correlation coefficient suggests that additional information on each of the method is needed to analyze the potential links between the AMS and volatility data. Moreover, Paciga et al. (2016) studied the volatility distribution of the PMF-derived organics and estimated that almost half of the SVOC, which was determined from PMF, is semi-volatile, while 42% is low-volatile and 6% is extremely low-volatile. This suggests that the two PMF-derived organic groups, commonly labeled for their oxidation levels, might not be directly linked to their actual volatilities.

[Figure]

Fig.1. Temperature profile along the axis of the heating section at a flow rate of 2 L/min.

List of changes:

Page 1, line 25-28: Text was changed to 'The volatility distribution of secondary organic aerosols formed and undergone aging, i.e. the particle mass fractions of semi-volatile, low-volatility and extremely low-volatility organic compounds in the particle phase was characterized in a boreal forest environment of Hyytiälä, Southern Finland.'

Page 1-2, line 47-51: Text was changed to: 'In general, the best agreement between the VTDMA results and the PMF-derived mass fractions of organics was obtained when $\Delta H_{VAP}$ = 80 kJ/mol was set for all organic groups in the model, with a linear correlation coefficient of around 0.4. However, this still indicates that only about 16% ($R^2$) of the variation can be explained by the linear regression between the results from these two methods.'

Page 2-3, line 91-99: Text was added as: 'Here, it needs to be noted that the volatility distribution of ambient aerosols does not represent the volatility distribution of the condensing organic compounds in the gaseous phase. However, it provides insights into the evaporation potentials of the compounds that are present in the particle phase. Furthermore, it will be useful for closure studies combining this information with condensation studies aiming to derive how the aerosol size distributions are affected by given gaseous species. Finally, measuring the evaporation of aerosols is also essential for testing the applicability and limitations of TD setups for inferring the volatility of aerosols.'

Page 3, line 105-108: Text was changed to: 'Typical organic groups determined using the PMF analysis include e.g. hydrocarbon-like OA (HOA), biomass burning OA (BBOA) and cooking OA (COA) or oxygenated OA (OOA).'

Page 3, line 120-125: Text was changed to: 'Typical organic groups determined using the PMF analysis include e.g. hydrocarbon-like OA (HOA), biomass burning OA (BBOA) and cooking OA (COA) or oxygenated OA (OOA).'

Page 4, line 157-159: Text was added as: 'The spread of the number size distribution of the aerosol was taken into account in the data inversion using the piecewise linear inversion approach (Gysel et al., 2009).'

Page 4, line 163-165: Text was added as: 'It was assumed that the particles were instantaneously thermally equilibrated with the surrounding gas phase, as the system was under atmospheric pressure.'

Page 4, line 167-175: Text was added as: 'The major particle losses during the heating process are from thermophoresis and Brownian diffusion (Wehner et al., 2002; Häkkinen et al., 2012). According to Ehn et al. (2007), who used a similar TD, the losses for aerosol particles above 15 nm in diameter were observed to be less than 20% when heated to 280 °C. Due to these losses, the VTDMA-measured data underestimates the mass concentration of the monodisperse aerosol particles after heating. However, this study was focusing on the change in

particle size, which should not be affected very much by the losses. Hence, the effect of the particle losses on the study results can be considered negligible.'

Page 4, line 182-184: Text was changed to: 'With VFR = 1 at a given temperature, particles are considered to not evaporate, while with VFR = 0 particles fully evaporate upon heating at that temperature.'

Page 6, line 260, line 262, 'Elemental carbon' was changed to 'EC'

Page 6, line 270-274, Text was changed to: 'This optimization method was constrained by setting the sum of mass fraction of organics from the model be equal to the mass fraction of OA measured by HR-AMS, and the mass fraction of each individual organic group to be larger than zero but lower than the total measured mass fraction of OA.'

Page 7, line 293-295, Text was changed to: 'This vaporization enthalpy ($\Delta H_{VAP}$) of Epstein et al. (2010) (Eq. 3) was also tested in the model calculations.'

Page 7, line 327-328, '$C^*$ dependent enthalpy' was changed to 'the $\Delta H_{VAP}$ of Epstein et al. (2010) '

Page 7, line 330-334: Text was changed to: 'In short, even though there have been afore-mentioned earlier studies reporting the $C^*$ and $\Delta H_{VAP}$ of AN and AS, we selected the ones shown by the red curves in Fig. 3 from our VTDMA technique for the model input to simulate the evaporation of ambient aerosols.'

Page 8, line 347-349: Text was added as: 'The different simulated evaporation behavior indicates that the model is sensitive towards $\Delta H_{VAP}$ values. '

Page 8, line 350-351: '$C^*$ dependent enthalpy' was changed to 'the $\Delta H_{VAP}$ of Epstein et al. (2010) '

Page 8, line 359-361: Text was changed to: 'By using the other vaporization enthalpy values (e.g. Combinations 1 to 8 in Table 2), better agreement between the fitted and observed thermograms (Fig. 3) was obtained. '

Page 8, line 365-368: Text was changed to: 'According to the performance of the model to TD data, the model was observed to be sensitive towards $\Delta H_{VAP}$ values. Low $\Delta H_{VAP}$ values (i.e., $\Delta H_{VAP}$ = 60-80 kJ/mol) are suggested to be used in the model in order to reproduce the measured thermograms.'

Page 8, line 378-380: Text was changed to: 'Since this study focuses on the volatility distribution of organics using a complex kinetic model, we chose to limit the PMF OA components to the main ones clearly connected with oxidation state.'

Page 9-10, line 421-434: Text was changed to: 'Using the enthalpy value of 60 kJ/mol for all organic groups, the modeled mass fraction of SVOA was higher than the SVOA from the PMF analysis. The opposite was true for LVOA, while

using $\Delta H_{VAP}$ values of 100 kJ/mol for all organic groups, the comparison results differed significantly from the 1:1 line.  With the enthalpy value of 80 kJ/mol for organics, the VTDMA-based OA composition was approximately equal to the ones from the PMF results, however, with a linear correlation coefficient of only 0.4. This relatively low correlation coefficient suggests that additional information on each of the method is needed for analyzing the potential links between the AMS and volatility data. Moreover, Paciga et al. (2016) studied the volatility distribution of the PMF-derived organics and estimated that almost half of the SVOC, which was determined from PMF, is semi-volatile, while 42% is low-volatile and 6% is extremely low-volatile.  This suggests that the two PMF-derived organic groups, commonly labeled for their oxidation levels, might not be directly linked to their actual volatilities.'

Page 10, line 455: '$C*$ dependent enthalpy' was changed to 'the $\Delta H_{VAP}$ of Epstein et al. (2010)'

Page 10, line 470-476: Text was added as: 'Tong et al. (2011) concluded that the diffusion coefficient of a viscous solution might affect the kinetics of evaporation of non-liquid particles, as aerosol particles in boreal forest environment are expected to be viscous according to Virtanen et al. (2010). Hence, also non-unity mass accommodation coefficients of a mixture and the particle-phase diffusion limitation on evaporation can add uncertainties to the interpretation of the TD data.'

Page 11, line 509-512: Text was changed to: 'Conclusively, these two case studies suggest that an effective $\Delta H_{VAP}$ value of 60-80 kJ/mol represent the boreal forest organic aerosols best and this effective $\Delta H_{VAP}$ value should be assumed in the model when comparing with the PMF results.'

Page 11, line 516-518: Text was changed to: 'The volatility of ambient aerosol particles formed and undergone aging was studied with a Volatility Tandem Differential Mobility Analyzer (VTDMA) in a boreal forest environment in Hyytiälä from April to May of 2014.'

Page 11, line 528-533: Text was changed to: 'The best correlation between the VTDMA results and the PMF-derived mass fractions of organics was obtained when $\Delta H_{VAP}$ = 80 kJ/mol was assumed for all organic groups in the model, with a linear correlation coefficient of around 0.4. This relatively low correlation coefficient indicates that we need to acquire additional information on each of the method to address the potential relation between the AMS and volatility data.'

Page 19: line 825-827: Text was added as: 'b: The particle mass concentration in particle size bin of 90-110 nm from DMPS is used to represent the particle mass concentration of the monodisperse aerosols (i.e. $D_P$ = 100 nm).'

Page 22: Axis was changed accordingly. Errorbar was added into the measurement dot.

Page 23: Measurement curve was changed to dot, while model result curves were changed to lines. Figure caption was changed to: 'Figure 3: An example of measured (black dots) vs. modeled (green, magenta and red lines) thermograms assuming different vaporization enthalpies of the organics.'

Page 25, line 916-918: Text was added as: 'Here, the Y-axis represents the VTDMA results interpretation using the kinetic model and the X-axis represents the AMS results interpretation using the statistical model (PMF).'

Page 26: Figure 6 in previous manuscript was moved to supplement, while previous figure 7 was changed to Fig. 6. Line 934: text was added as: '$\Delta H_{vap} = 80$ kJ/mol was used in the kinetic evaporation model.' Legend of the figure was changed to: 'Kinetic model results' on the left, 'Statistical model results (PMF) on the right'.

Marked-up Manuscript:

[revised manuscript text omitted]

that are present in the particle phase. Furthermore, it will be useful for closure studies combining this information with condensation studies aiming to derive how the aerosol size distributions are affected by given gaseous species. Finally, measuring the evaporation of aerosols is also essential for testing the applicability and limitations of TD setups for inferring the volatility of aerosols.

Positive Matrix Factorization (PMF) is one of the widely used factor analysis techniques for environmental applications. PMF allows separating organic aerosol (OA) mass spectra into individual groups based on their bulk chemical characteristics, providing information on the OA sources and atmospheric processing (Lanz et al., 2007; Huffman et al., 2009; Zhang et al., 2011). Typical organic groups determined using the PMF analysis include e.g. hydrocarbon-like OA (HOA), biomass burning OA (BBOA) and cooking OA (COA) or oxygenated OA (OOA). OOA can be further separated into low volatility OOA (LV-OOA) and semi-volatile OOA (SV-OOA). Even though there have been multiple studies using PMF to identify different organic OA groups from ambient data (Ulbrich et al., 2009; Hildebrandt et al., 2010; Ng et al., 2010), especially the SV-OOA and LV-OOA groups, to our knowledge there are only few studies (Cappa and Jimenez, 2010; Paciga et al., 2016) attempting to directly connect the oxygenation levels from these two OOA groups with the volatility of OA obtained by other methods. Comparing the volatility distribution obtained using a mass transfer model and VTDMA data to the oxidation level derived from the AMS data using PMF can help in quantifying the volatilities of SV-OOA and LV-OOA.

In this study, we provide quantitative information on volatility distributions of organic species of ambient aerosol in a boreal forest environment. The sensitivity of the kinetic model was tested towards different parameters of organic compounds, including density, molar mass, saturation vapor concentration, diffusion coefficient and vaporization enthalpy values. More specifically, the sensitivity result to assumed vaporization enthalpy values of organics is discussed. The VTDMA-derived volatility distributions are compared with the ones obtained from the statistical analysis of the AMS.

2 Methods

2.1 Measurements site

The measurements were performed at the Hyytiälä SMEAR II (Station for Measuring Ecosystem-Atmosphere Relations II) between 14 April and 31May 2014. The SMEAR II station, located in Southern Finland, is surrounded by a 54-year-old pine forest. The closest large city is Tampere with a population of around 213 000 and about 48 km to the South-West of the measurement station.

A series of ambient parameters, e.g., particle number size distribution of 3-1000 nm particles (Aalto et al., 2001), ambient meteorological conditions such as temperature, relative humidity, solar radiation, wind speed and wind direction as well as gas phase concentrations of e.g. $SO_2$, $O_3$, $NO_X$, are continuously measured at the station.

**2.2 Particle Volatility**

The evaporation behavior of submicron aerosols was investigated using a Volatility Tandem Differential Mobility Analyzer (VTDMA), which is part of a Volatility-Hygroscopicity Tandem Differential Mobility Analyzer (VH-TDMA) system (Hong et al., 2014). A brief schematic view of the VTDMA is shown in Fig. 1. In brief, a monodisperse aerosol population (particle diameter of 30, 60, 100 and 145 nm; RH < 10%) was selected by a Hauke-type Differential Mobility Analyzer (DMA; Winklmayr et al., 1991). The aerosol flow was then heated by a thermodenuder at a set temperature, after which the remaining aerosol material was introduced into a second DMA followed by a condensation particle counter (CPC, TSI 3010 & TSI 3772), where the number size distribution of the aerosol after heating was measured. The spread of the number size distribution of the aerosol was taken into account in the data inversion using the piecewise linear inversion approach (Gysel et al., 2009). The thermodenuder is a 50-cm stainless steel tube. No adsorptive material for removing the gas phase was used after the heating section. The residence time inside the thermodenuder was around 2.5 s. The heating temperature of the setup ramped from 25 °C to 280 °C with a time resolution of about an hour. It was assumed that the particles were instantaneously thermally equilibrated with the surrounding gas phase, as the system was under atmospheric pressure.

The major particle losses during the heating process are from thermophoresis and Brownian diffusion (Wehner et al., 2002; Häkkinen et al., 2012). According to Ehn et al. (2007), who used a similar TD, the losses for aerosol particles above 15 nm in diameter were observed to be less than 20% when heated to 280 °C. Due to these losses, the VTDMA-measured data underestimates the mass concentration of the monodisperse aerosol particles after heating. However, this study was focusing on the change in particle size, which should not be affected very much by the losses. Hence, the effect of the particle losses on the study results can be considered negligible.

The VTDMA measures the particle diameter (and concentration) after heating at each temperature for particles of certain initial size. From this information volume fraction remaining (VFR) after the heating of particles of diameter $D_P$ can be defined as follows

$$VFR(D_P) = \frac{D_p{}^3(T)}{D_p{}^3(T_{room})} = GF_V^3(T).\qquad(1)$$

$GF_V$ describes how much of the particles shrink in size upon heating. With VFR = 1 at a given temperature, particles are considered to not evaporate, while with VFR = 0 particles fully evaporate upon heating at that temperature. The mass fraction remaining (MFR) after the heating was assumed to be equivalent to VFR assuming that particle density was constant upon heating (Häkkinen et al., 2012).

[revised manuscript text omitted]

---

## Referee Report (RR1)

Review of acp-2016-774
Hong et al. "Estimates of organic aerosol volatility in a boreal forest using two independent methods"

**Overview:**
I have reviewed this manuscript twice: once as a Quick Access Review for the initial submission and once for Referee Comments. The authors have sufficiently addressed all of my concerns and modified all of their discussion accordingly. I recommend that this manuscript be accepted as is.